# Hydrophilization of Hydrophobic Mesoporous High-Density Polyethylene Membranes via Ozonation

**DOI:** 10.3390/membranes12080733

**Published:** 2022-07-26

**Authors:** Polina M. Tyubaeva, Mikhail A. Tyubaev, Vyacheslav V. Podmasterev, Anastasia V. Bolshakova, Olga V. Arzhakova

**Affiliations:** 1Academic Department of Innovative Materials and Technologies, Plekhanov Russian University of Economics, Stremyanny Per. 36, 117997 Moscow, Russia; tyubaev.ma@rea.ru; 2Department of Biological and Chemical Physics of Polymers, Emanuel Institute of Biochemical Physics, Russian Academy of Sciences, ul.Kosygina 4, 119334 Moscow, Russia; vpodmasterev@yandex.ru; 3Faculty of Chemistry, Lomonosov Moscow State University, Leninskiye Gory 1/3, 119991 Moscow, Russia; nastya@polly.phys.msu.ru (A.V.B.); arzhakova8888@gmail.com (O.V.A.)

**Keywords:** mesoporous membanes, high-density polyethylene, ozonation, hydrophilization, surface modification

## Abstract

This work addresses hydrophilization of hydrophobic mesoporous membranes based on high-density polyethylene (HDPE) via ozonation. Mesoporous HDPE membranes were prepared by intercrystallite environmental crazing. Porosity was 50%, and pore dimensions were below 10 nm. Contact angle of mesoporous membranes increases from 96° (pristine HDPE) to 120° due to the formation of nano/microscale surface relief and enhanced surface roughness. The membranes are impermeable to water (water entry threshold is 250 bar). The prepared membranes were exposed to ozonation and showed a high ozone uptake. After ozonation, the membranes were studied by different physicochemical methods, including DSC, AFM, FTIR spectroscopy, etc. Due to ozonation, wettability of the membranes was improved: their contact angle decreased from 120° down to 60°, and they became permeable to water. AFM micrographs revealed a marked smoothening of the surface relief, and the FTIR spectra indicated the development of new functionalities due to ozonolysis. Both factors contribute to hydrophilization and water permeability of the ozonated HDPE membranes. Hence, ozonation was proved to be a facile and efficient instrument for surface modification of hydrophobic mesoporous HDPE membranes and can also provide their efficient sterilization for biomedical purposes and water treatment.

## 1. Introduction

The challenges of the 21stcentury demand searches for innovative polymeric materials with task-oriented properties and functionalities for their practical use in science, technology, and daily life. The development of new materials can be conditionally divided into two general lines: synthesis of new polymers with desired characteristics and modification of existing polymers. The advantage of the second strategy involves the use of high-tonnage commercial polymers including polyolefins, among which polyethylenes have the biggest annular global production in 2021 (107 million metric tons worldwide): by 2029, the global market volume of polyethylene is expected to rise to around 130 million metric tons [1]. Surface modification offers an advantageous avenue for controlled variations in the performance of various polymers and polymer-based membrane materials. These methods involve plasma treatment, irradiation with gamma-rays, corona discharge, UV irradiation, ion beam treatment, etc. [2,3,4,5,6,7,8]. Among the above-mentioned methods for surface modification of polymers, ozonation has captured the special attention of scientists and engineers, as molecular ozone (an allotropic modification of oxygen containing three atoms of oxygen) is a powerful natural oxidant with an oxidation potential of 2.07 V in gaseous phase [9,10,11,12]. Ozone is widely used in daily practice for sterilization of water, bleaching of wood pulp, disinfection, as a pesticide in agriculture, animal husbandry and fish farming, etc. [13].

The advantages of ozonation for surface modification of polymers [14,15,16,17,18,19] are primarily concerned with the high oxidative potency of ozone, available equipment, low cost, natural origin of ozone, ecological safety, efficient and controlled regimes in the reactor, and ability to treat the samples with complex surface morphology and complicated shapes [19,20]. In addition to the solution of practical issues, studies in this direction are deeply connected with scientific problems which involve the search for new strategies which extend the benefits of this approach and allow preparation of innovative polymeric materials with task-oriented properties as well as the assessment of the potential of ozonation for degradation of polymers and solution of ecological challenges of waste treatment. The problems of hydrophilicity of initially hydrophobic polymers can be also solved by ozonation of polyolefins (PE, PP) coupled to ozone-induced chemical reactions including graft polymerization of hydrophilic monomers such as acrylamide [16,21]. Nowadays, numerous scientific studies are devoted to surface hydrophilization of diverse polymers, especially hydrophobic polymers, as this transformation can substantially broaden the areas of their practical applications as membranes, scaffolds for tissue engineering, and sorbents. This goal can be achieved by different ways: grafting, photografting, ink-jet printing, surface charging, deposition of hydrophilic coating, etc. [22,23,24,25,26,27]. 

This work addresses the preparation of mesoporous membranes based on highly hydrophobic high-density polyethylene (HDPE) with nanoscale pore dimensions (below 10 nm), characterization of their morphology and performance as membrane materials, their further treatment by ozonation, and assessment of this post-treatment as an efficient method for modification and hydrophilization of hydrophobic membrane materials. 

## 2. Materials and Methods

### 2.1. Materials

In this study, we used the films of HDPE (Figure 1) (DSM, Sittard, The Netherlands); the film thickness was 25 µm; M_n_~140,000. According to the DSC tests, the degree of crystallinity was estimated from the heat of fusion to be equal to 59%. The amorphous phase of the HDPE films exists in the rubbery state (glass transition temperature is −70 °C). 

As a physically active solvent, we used *n*-decane (Sigma-Aldrich, Burlington, MA, USA). 

### 2.2. Preparation of Mesoporous Membrane Materials

Porous membranes were prepared according to the following procedure: initial HDPE films with a gage size of 5 × 3 cm were fixed in hand-operating clamps. Then, the samples were stretched to the desired tensile strain *ε* in the presence of *n*-decane. Then, the samples under isometric conditions were dried under pressurized air and annealed at 110 °C. The resultant samples were characterized by high shape stability even at elevated temperature and upon treatment with organic solvents. 

**Porosity.** Porosity *W* of the HDPE films upon stretching in *n*-decane was estimated from changes in the geometric dimensions of the samples with increasing the tensile strain *ε*. Thickness of the samples was measured using an IZV-2 optimeter (Metronom, Moscow, Russia). The volume porosity *W* was calculated according to the following equation: (1)$$W=\frac{\Delta V}{V_{0}+\Delta V}\times 100\%$$
where *V*_0_ is the initial volume of the sample, Δ*V* is the difference between the volume of the deformed sample and initial sample (upon tensile drawing). The measurements were performed for 3–5 samples. The experimental error was less than 3%.

### 2.3. Methods

#### 2.3.1. Differential Scanning Calorimetry

Thermal properties of the PE samples were studied using a Netzsch 214 Polyma thermal analyzer (Selb, Germany); the tests were performed in air; heating rate was 10 K/min; the cooling rate was 10 K/min. The DSC tests in air were identical to the DSC scans in argon due to low oxidation of the samples. The weight of the test samples was 6–7 mg. The temperature interval was varied from 20 °C to 220 °C. The average statistical error was ±2.5%.

Heat of fusion, Δ*H*, was calculated using the NETZSCH Proteus software according to the standard technique [28].

Degree of crystallinity, *χ*, was calculated from the area of the melting peak as:(2)$$\chi =\frac{\Delta H}{H_{PE}}\times 100\%$$
where ∆*H* is the heat of fusion; *H_PE_* is the heat of fusion of an “ideal” crystal of PE (293 J/g).

Lamellar thickness  l was calculated according to [29]: (3)$$T_{m}=T^{0}{}_{m}\;\left(1-\frac{2\gamma }{\Delta H\times l}\right)$$
where *T*^0^*_m_* is the equilibrium melting temperature of an ideal PE crystal [30], γ is the top and bottom fold surface free energy, *l* is the lamellar thickness, *T_m_* is the experimental melting temperature, and Δ*H* is the heat of fusion per cubic centimeter of the perfect crystal. γ = 0.09 J m^−2^ for PE [30].

#### 2.3.2. Ozonation

The test samples were ozonated using a flow-through reactor (Medozone, Moscow, Russia). Ozone was synthesized from oxygen in a barrier discharge. The required ozone concentration in the in-flux tent was set by voltage at ozonation apparatus electrodes (5–7 kV). Ozone concentration was controlled spectrophotometrically at a wavelength of 254 nm. The basic working ozone concentration was 5.5 × 10^−5^ mol/L. The duration of ozonation was varied from 1 to 25,200 s. The rate of ozone flow was 0.1 L/min. Ozone uptake was calculated as the difference between input and output ozone content (in the presence of the sample in the reactor), respectively.

#### 2.3.3. Fouirier Transform IR-Spectroscopy 

FTIR spectra of samples of the material were collected on a Lumos BRUKER instrument (Karlsruhe, Germany) by the method of multiple disturbed total internal reflection on a diamond crystal. The resolution was 2 cm^−1^. Measurements were carried out in the range from 600 to 4000 cm^−1^. Conclusions about changes in the composition of the surface layer of the polymer material were made based on the analysis of absorption bands [31].

#### 2.3.4. Atomic Force Microscopy (AFM)

Surface morphology of the test samples was studied by atomic force microscopy (AFM) on a Multimode microscope with a Nanoscope V controller (Veeco, Plainview, New York, NY, USA). The atomic force microscopy (AFM) observations were performed in air at room temperature under the tapping mode. As probes, high aspect ratio polysilicon cantilevers (TipsNano, Tallinn, Estonia) were used; the nominal resonance frequency was 125 kHz, the force constant was 3.5 N m^−1^, and Q-factor was about 280. Image processing was performed using FemtoScan Online software (Advanced Technologies Center, Moscow, Russia) [32].

#### 2.3.5. Pressure-Driven Liquid Permeability Method for Water Flow Estimation 

The HDPE samples before and after ozonation were studied by the method of pressure-driven liquid permeability (PDLP). The samples were placed into an FMO-2 membrane cell filled with water. The pressure gradient was 1 bar. The measurements of the water flow rate *Q* were performed for not less than 5–7 samples. The experimental error was below 3–5%. Average pore diameter *D_p_* (pore radius *r*) was calculated within the Hagen–Poiseuille approximation and liquid flow rate *Q* through the porous structure as a network of open narrow channels reads as
(4)$$Q=\frac{\pi nr^{4}S\Delta p}{8\eta d}=\frac{Wr^{2}S\Delta p}{8\eta d}$$
where *n* is the number of pores per unit surface, *r* is the pore radius, Δ*P* is the pressure gradient, *d* is the thickness of the film, *η* is the viscosity of the liquid, *S* is the membrane area.

#### 2.3.6. Mechanical Tests

Mechanical properties of the test samples were examined on a Devotrans DVT GP UG 5 universal tensile machine (Istanbul, Turkey). The strain rate was 25 mm/min. The size of the samples was 10 × 40 mm. The results were averaged for 5–7 samples. Tensile strength was estimated using the Devotrans software. The average statistical error was ±0.02 MPa. Elongation at break *ε* was calculated as:(5)$$\varepsilon =\frac{\Delta l}{l_{0}}\times 100\%$$
where ∆*l* is the difference between the final and initial length of the sample; *l*_0_ is the initial length of the sample. The average statistical error was ±0.2%.

#### 2.3.7. Water Contact Angle Measurements

Wettability of the test samples before and after ozonation was evaluated by the water contact angle (WCA) by the sessile drop method. Water droplets (5 µL) (deionized water) were placed onto three different areas of the membrane surface using an automatic dispenser. The contact angle of the tests samples was measured using an M9 No. 63649 lens FMA050 optical microscope (Moscow, Russia). Image processing was performed using Altami studio 3.4 software. The results of three measurements from different regions of the sample were averaged. The experimental error was ±0.5%.

## 3. Results

This work is focused on the advantages of ozonation for structural modification of hydrophobic mesoporous (MP) HDPE membranes prepared by the strategy of environmental crazing and involves controlled preparation and characterization of the mesoporous membranes, selection of optimal membranes for ozonation, and investigation of the effect of ozonation on the performance of ozone-treated membranes, including wettability, water contact angle, water flux, and mechanical characteristics.

### 3.1. Mesoporous Membranes Based on High-Density Polyethylene: Preparation and Performance: Preparation, Structure, and Performance

Environmental crazing is known to be the specific mode of plastic deformation of semicrystalline and amorphous glassy polymers, which is accompanied by the development of macroscopic porosity with nanoscale pore dimensions (below 10 nm) [33,34,35,36]. In the case of semicrystalline polymers, their deformation in the presence of physically active liquid environments (PALE) proceeds via the mechanism of intercrystallite crazing which involves stress-induced cavitation and fibrillation in the amorphous phase upon separation of crystalline lamellae [37,38,39]. In this study, the porous membranes were prepared by tensile drawing of pristine HDPE films in the presence of the PALE to different tensile strains. As the PALE, *n*-decane was selected according to the following criteria: this organic solvent is characterized by a high affinity towards hydrophobic HDPE, equilibrium degree of swelling of *n*-decane in HDPE is equal to 10 wt.%, and its Hildebrand solubility parameter is close to that of HDPE (19.00 and 16 (J/cm^3^)^0.5^ for HDPE and *n*-decane, respectively) [40]. Upon tensile drawing in *n*-decane, porosity *W* of the samples gradually increases with increasing tensile strain *ε,* and at *ε =* 200%, W = 50 vol.%. After stretching, the porous samples were characterized by high strain recovery, and upon unloading, they resumed their initial dimensions. This problem of low shape stability was solved when the samples were annealed under isometric conditions. The optimal temperature of annealing *T_an_* was estimated from thermomechanical tests, and *T_an_* = 110 °C (by 20 °C below the melting temperature of HDPE). The resultant samples show excellent shape stability even at elevated temperatures up to 80 °C. Pore dimensions *D_p_* of the prepared stable materials were estimated by the methods of permporometry, pressure-driven liquid permeability, and low-temperature nitrogen adsorption. All estimated show good correlation; *D_p_* = 8 nm. Hence, these materials can be classified as mesoporous materials according to the IUPAC classification [41]. Figure 2 shows the details of the inner morphology of the as-prepared membranes.

The inner structure of the mesoporous (MP) HDPE membranes is composed of crystalline lamellae and fibrils within the intercrystalline space. The fibrils are oriented along the direction of tensile drawing and separated by elongated pores. The diameter of fibrils is equal to 7 nm, and the pore dimensions are equal to 8 nm. Water contact angle (WCA) of the MP membrane increases as compared with pristine HDPE and appears to be equal to 120°; in other words, WCA increases by ~25°. This fact indicates that the MP membranes are characterized by high hydrophobicity. The above membranes were tested with respect to water permeability, and at low pressure gradient (below 5 bar), they appear to be impermeable to water (water flux = 0). The water entry pressure of the MP membranes was estimated according to the well-known Laplace–Washburn equation: (6)$$P=\frac{2\gamma }{r}\cos\theta$$
where *P* is the water entry pressure, *γ* is the surface tension of a liquid (water), *r* is the pore radius, *θ* is the contact angle. Contact angle for the water/MP HDPE system is *θ* = 120°; the surface tension of water is typically 0.07197 N/m at 25 °C. Hence, the water entry pressure is very high and equal to *P* = ~250 bar. Therefore, the as-prepared MP membrane materials are characterized by high porosity and high hydrostatic resistance. This marked enhancement of hydrophobicity can be explained by the increased surface roughness within the Cassie–Baxter theory [42].

According to the analysis of the AFM images, surface roughness of the MP membranes is ~6 times higher than that of the pristine HDPE. The profiles of the MP samples along the tensile drawing direction and in the perpendicular direction (Figure 2C,D) reveal the presence of both nanoscale and microscale surface reliefs. As a result, the MP HDPE membranes are characterized by high hydrohobicity due to enhanced surface roughness. Hence, our further studies are directed towards surface modification of the MP HDPE membranes by ozonation. 

### 3.2. Ozonation of HDPE Mesoporous Membranes 

The test samples of MP membranes were ozonated in the flow reactor at room temperature (+22 °C) when the concentration of ozone in the ozone-oxygen flow was varied from 2.2 × 10^−4^ to 4.2 × 10^−3^ mol/L and the gas flow rate was changed from 3.3 × 10^−3^ to 5.0 × 10^−2^ L/s. Ozone was synthesized from oxygen in the barrier discharge, and concentration of ozone was measured in the off-gas flow, thus making it possible to estimate the ozone uptake by the test samples as
Ozone uptake (%) = [O_3_,_Blank_] − [O_3_,_MP HDPE_]/O_3_,_Blank_ × 100

In the case of ozonation of polyethylenes, the reaction commences at the surface and propagates inward the volume via ozone diffusion. In this work, to achieve marked structural surface modification of the samples, two routes are possible: prolong treatment (dozens of hours) with low ozone concentrations (~5.5 × 10^−6^ mol/L) or short-term treatment with high ozone dosage (~10^−3^ mol/L). According to data in the literature, the second route has been proved to be the most reliable [15]. Figure 3 presents the kinetic curves of ozone uptake by MP HDPE membrane in comparison with the blank test and pristine HDPE. 

Ozone uptake of pristine HDPE is low and nearly the same as in the case of the bland test (without the sample). At the same time, mesoporous HDPE membrane is characterized by high ozone uptake. The results on ozone uptake are listed in Table 1. Along with increasing the duration of ozonation, ozone uptake increases from 4.2 × 10^−9^ to 5.6 × 10^−9^.

In general, the process of ozone uptake can be conditionally divided into two stages: high-rate stage and low-rate stage in the steady-state regime [43]. The duration of the high-rate stage is equal to 70 s for pristine HDPE and 150 s for the MP membrane. At this stage, ozone uptake is maximal. Due to the highly developed surface of the MP membrane, ozone uptake is six times higher than that of the non-porous pristine HDPE. Let us mention that, when ozonation time increases to 25,200 s, the membrane becomes brittle and easily loses its integrity. In some ways, ozonation may be considered as a powerful instrument for degradation of plastics. 

### 3.3. The Effect of Ozonation on Structure and Performance of Mesoporous HDPE Membranes

In this section, we will discuss the structure and morphology of the ozonated membranes, their performance, and the benefits of ozonation as the method for structural modification of hydrophobic membranes.

#### 3.3.1. Fourier Transform IR Spectroscopy for Ozonated Mesoporous HDPE Membranes

The effect of ozonation on the test samples was analyzed by FTIR (Lumos, Bruker, Germany) in the wavenumber ranging from 4000 cm^−1^ to 400 cm^−1^ with a resolution of 2 cm^−1^. Figure 4 shows the corresponding FTIR spectra for pristine HDPE and MP membrane after ozonation.

The FTIR spectra show the presence of new functionalities (OH, C=O, etc.) in the ozonated membrane as compared with the spectrum of pristine HDPE.

#### 3.3.2. Differential Scanning Calorimetry

Differential scanning calorimerty (DSC) is credited as the most reliable method for the description of phase composition of the samples and allows us to reveal the effect of ozonation on the heat of fusion and degree of crystallinity of the test samples. Figure 5 shows the corresponding DSC scans for pristine HDPE, mesoporous HDPE membranes, and ozonated HDPE membranes.

The results of the DSC tests as the data on melting temperature, heat of fusion, and degree of crystallinity are listed in Table 2.

According to the DSC tests, when the duration of ozonation is low (600s), melting temperature of the initial and ozonated membranes remains unchanged. As the duration of ozonation increases to 25,200 s (so-called “severe” ozonation), melting temperature slightly decreases, and the onset of melting is shifted to low temperatures (Figure 5). During the ozonation for 10 min, heat of fusion decreases by 15%, which means that the degree of crystallinity decreases by ~8%. However, when the duration of ozonation increases to 25,200 s, heat of fusion markedly increases, and the degree of crystallinity appears to be equal to 90.5%. The polydispersity of crystallites in the samples can be characterized by the half width of the melting peak. As follows from Figure 5, this parameter markedly increases upon ozonation, and for the samples under “severe” ozonation, this value attests that ozone-induced oxidation provides recrystallization and formation of crystallites with high polydispersity. The final material becomes fully crystalline and exceptionally brittle (when ground by one’s fingers, it disintegrates into powder with micronic dimensions of the particles). Moreover, this fact means that oxidation primarily affects the amorphous phase and leads to chain scission and formation of crystallizable oligomers. Hence, the sufficient duration of ozonation is 600 s when the treated membranes acquire the hydrophilicity but preserve their integrity and show good mechanical properties. 

#### 3.3.3. Water Contact Angle

Wettability of ozonated samples was studied via the sessile drop method with respect to deionized water at room temperature, and water contact angle (WCA) was measured. Figure 6 shows the diagram illustrating changes in WCA from pristine HDPE to mesoporous membranes and, finally, to ozonated membranes. 

As compared with pristine HDPE, WCA of mesoporous membranes increases from 96° to 120° due to increased surface roughness. As a result of ozonation, WCA decreases from 120° down to 60°. This fact implies that WCA of the ozonated membranes is even lower than that of pristine HDPE. According to the adopted classification, the borderline between hydrophobic and hydrophilic materials corresponds to 90°. Hence, the ozonated material can be considered to be hydrophilic as opposed to traditional hydrophobic HDPE [44].

#### 3.3.4. Water Permeability

Ozonated HDPE membranes were studied by the method of pressure-driven liquid permeability with respect to the permeation of pure water. Let us restate that the initial MP membranes were waterproof and the water permeability threshold was equal to 250 bar. Ozonated membranes appeared to be permeable to water, and the water flow was equal to 2 L/(m^2^ h) at a pressure gradient of 1 bar. This fact suggests that ozonation provides a marked hydrophilization of the initially highly hydrophobic membranes.

#### 3.3.5. Mechanical Tests

High mechanical properties are known to be one of the key characteristics of the membrane materials that controls their performance and lifetime. The results of tensile tests are listed in Table 3.

The data on mechanical tests show that, even after ozonation for 600 s, the ozonated HDPE membranes preserve their high mechanical strength and deformability, thus suggesting that this mode of treatment can be recommended for practical use. Let us mention that prolonged ozonation leads to complete degradation and disintegration of the membranes (‘severe” ozonation). 

#### 3.3.6. AFM Observations

Atomic force microscopy is credited as the powerful instrument for the characterization of structure and morphology of the porous membrane materials. Figure 7 shows the AFM micrographs of the HDPE membranes after ozonation.

As can be seen in the AFM micrographs and corresponding profilograms along the direction of tensile drawing and in the transverse direction (Figure 7), the surface relief of the ozonated membranes appears to be more flattened and less pronounced. This fact may be invoked to explain the lower water contact angle of the membranes by the reduced surface roughness according to the Cassie–Baxter theory. 

## 4. Discussion

Ozone is known to be a powerful oxidant for various chemicals, including hydrocarbons and polymers (PE, PP).Its oxidation reactions and their kinetics and reaction products have been extensively discussed in the literature [20]. Ozone is widely used for water treatment, food processing, preservation, microbial inactivation, and modification of different materials, including textiles, etc. [20,45,46]. Ozonation also provides marked structural changes in diverse polymers, thus leading to the modification of their structure and properties [15,20]. 

Controlled modification of membrane materials, including commercial membranes, offers new advantages for their practical use in various applications such as ultrafiltration, nanofiltration, membrane contactors, water purifications systems, etc. This work is focused on the potential of ozonation for modification of mesoporous membrane materials based on HDPE prepared by the platform of environmental intercrystallite crazing. This strategy allows target-oriented preparation of HDPE membranes with high porosity and pore dimensions below 10 nm (Figure 2). The inner porous structure of the resultant materials is composed of thin crystallites and fibrillated amorphous phase in the intercrystalline regions. This structure is produced due to cavitation and fibrillation of the soft amorphous phase upon separation of crystallites, upon tensile drawing in the presence of physically active liquid environments via the Raleigh–Taylor meniscus instability mechanism. These membranes materials are characterized by high surface roughness due to the formation of micro/nanostructured surface relief and high WCA = 120° (Figure 2). As a result, these membranes are highly hydrophobic and impermeable to water as their water entry pressure is 250 bar. In this work, we studied the effect of ozonation on the performance of the mesoporous HDPE membranes when the samples are exposed to ozone treatment in the reactor at room temperature. 

Ozonation of mesoporous HDPE membranes is accompanied by a marked ozone uptake as compared with that of pristine non-porous HDPE, and this value increases from 0.8 × 10^−9^ to 4.2 × 10^−9^ L (exposure for 600 s) and from 0.9 × 10^−9^ 5.6 × 10^−9^ L (Figure 3). This increased ozone uptake can be explained by the well-developed porosity of the test membranes, and ozone can easily penetrate into the mesoporous structure and attack both amorphous and crystalline phases. The analysis of the FTIR spectra (Figure 4) of the ozonated samples reveal the presence of new functionalities in the membranes 1715 cm^–1^ (OH, C=O, C-O, etc.). Hence, as a result of ozone treatment, the chemical composition of the membranes has changed, and new functionalities can lead to predictable variations in the performance of the ozonated membranes. 

According to the DSC tests (Figure 5), ozonation leads to marked changes in the phase composition of the test membranes. Upon short-term ozonation, melting temperature remains virtually unchanged but the enthalpy of fusion decreases, and the degree of crystallinity is reduced by 10% (Table 1). This fact implies that the polymer experiences a partial amorphization, and this evidence agrees with the literature data on ozonation of oriented PE [20]. Ozone attacks the tertiary C-H bonds in both amorphous and crystalline regions, and the number of defects in the crystalline phase increases, thus facilitating the ozone access. In this way, crystallites are partially destroyed and transformed to the amorphous phase. This ozone-assisted degradation of crystallites is also enhanced by the porous structure of the MP membranes when ozone is able to penetrate into the volume and attack crystallites at the folded surface. Moreover, the melting peak becomes wider, and the onset of melting is shifted to lower temperatures (Figure 5). The half-width of the melting peak can serve as the measure of polydispersity of crystallites. This evidence suggests that ozone-induced oxidation leads to the partial disruption of crystalline phase, thus leading to the formation of small-sized crystallites. Upon prolonged exposure, enthalpy of fusion markedly increases, the degree of crystallinity appears to be equal to 90.5%, and the melting peak appears to be wide (Figure 5). The final fully crystalline material becomes brittle and disintegrates into powder. This behavior can be explained by the degradation of amorphous phase, chain scission, formation of short-chained oligomers and low-molecular-mass products, and recrystallization. In principle, prolonged ozonation can be used for degradation and utilization of polyolefins, but this problem requires further studies and detailed characterization of degradation kinetics supported by the GPC analysis of degradation products, which is beyond the scope of this article. Hence, for surface modification of the MP HDPE membranes, the sufficient exposure for ozonation is 600 s when the membranes preserve their integrity and show good mechanical properties. 

Let us consider the performance of the ozonated membranes. The measurements of water contact angle (WCA) provide an important information concerning the wettability of the membrane materials as the measure of interaction between the polymer material and water. This parameter also serves as a measure of hydrophobicity or hydrophilicity of the material: when WCA < 90°, the material is said to be hydrophilic, and at WCA > 90° -hydrophobic [44]. For pristine HDPE, WCA = 96° and this polymer is traditionally known to be hydrophobic. However, after environmental intercrystallite crazing, the resultant MP HDPE membrane materials have WCA = 120°, thus indicating that their hydrophobicity is markedly increased due to the formation of micro/nano-patterned surface relief in full accordance with the Cassie–Baxter theory (Figure 6). According to the image analysis of the corresponding AFM micrographs (Figure 2), their surface roughness is six times higher. After ozonation, WCA of the ozonated materials decreases down to 60°, which is by 36° lower than that of pristine HDPE (Figure 6). Hence, the ozonated materials can classified as hydrophilic. The reasons behind this transition from hydrophobic to hydrophilic character can be explained by structural changes of the membranes. Image analysis of the AFM micrographs and profilograms of the ozonated membranes (Figure 7) shows that, upon ozonation, the surface patterned relief becomes more flattened, and the difference between the heights of surface pattern is reduced. The second reason of hydrophilization is concerned with changes in the chemical composition of the ozonated HDPE membranes as indicated by the FTIR measurements. The presence of new functionalities also noticeably contributes to the improved hydrophilicity.

As a result of improved hydrophilicity, the ozonated membranes become permeable to water. Let us mention that initial MP membranes are waterproof. Due to ozonation, water flux increases from zero to 2 L/(m^2^ × h) at 1 bar. This effect is preserved during repeated tests (for 2 months). Pore radius *r* of the ozonated membranes was estimated from the water flux according to the Hagen–Poiseuille equation for liquid flow through porous media (Equation (4)): *r* = 4 nm, which is equal to that of the initial MP membranes. Upon ozonation, the membranes preserve their shape stability and porosity. The mechanical properties of the membrane materials upon mild ozonation (600 s) are also preserved at the level of initial MP materials. Hence, ozonation can be considered as a safe and facile method for the hydrophilization of initially hydrophobic HDPE membranes. Moreover, ozonation has been shown to bean efficient instrument for bacterial inactivation for a wide spectrum of microorganisms, and ozone treatment under mild conditions can provide both sterilization and hydrophilization of the membrane materials, which is important for biomedical purposes and fine water treatment in optoelectronics. 

## 5. Conclusions

Surface modification of polymeric membranes presents an important direction for modern polymer and materials science and offers new advantages for their use in different applications, including biomedical purposes, fine water treatment, optoelectronics, etc. This work reports the results concerning preparation of mesoporous polymeric membranes based on commercial HDPE films via the mechanism of environmental intercrystallite crazing and their further modification by ozone treatment in the reactor. This approach allows preparation of the mesoporous HDPE membranes (thickness 17 µm) with a porosity of 50% and nanoscale pore dimensions (below 10 nm). The as-prepared membranes are characterized by increased hydrophobicity due to the development of micro/nanostructured surface and enhanced surface roughness (by six times higher than that of pristine HDPE), and their contact angles increase from 96° (pristine HDPE) to 120°. Hence, these membrane materials are highly hydrophobic and can be used in practice as lyophobic materials for membrane separation, selective sorption, as waterproof breathable air/gas-permeable materials, etc. Due to well-developed hydrophobicity provided by chemical nature of HDPE and enhanced surface roughness, the mesoporous membranes are impermeable to water and aqueous solutions (for example, aqueous solutions of biological species), and their water entry threshold is high and equal to 250 bar. Ozonation of the hydrophobic MP membranes in the flow reactor is accompanied by a marked ozone uptake, proceeds via surface and bulk oxidation of HDPE, and leads to their marked hydrophilization. As a result of ozonation, the water contact angle of MP membranes decreases from of 120° down to 60°. According to the adopted classification, the resultant materials can be rated as hydrophilic as their contact angle is below 90°. The improved hydrophilicity is provided by reduced surface roughness of the ozonated membranes and formation of new functionalities on their surface and in the bulk. Due to ozonation, the ozonated membranes become permeable to water, and the water flux is equal to 2 L/(m^2^ × h) at 1 bar. Mild short-term ozonation can be considered as the leads to the modification of the membrane materials as they preserve their shape stability and show good mechanical properties. Upon “severe” long-term ozonation, the HDPE membranes become exceptionally brittle and disintegrate into powder. This method of “severe” ozonation can be recommended for efficient degradation of the HDPE membranes as intensive ozonolysis provides chain scission and formation of oligomer and low-molecular-mass products, and our further studies will be focused on the detailed examination on the kinetics of ozone-assisted oxidation. The proposed approach offers a facile and efficient method for controlled modification of HDPE membranes and allows their transformation from hydrophobic to hydrophilic materials. As a result, the areas of their practical use can be substantially widened from fine water treatment to biomedical applications (for example, membranes for nanofiltration, biocompatible-materials, and scaffolds for tissue engineering). Moreover, ozonation under controlled conditions can be used for sterilization of the membranes, thus solving both problems of hydrophilization and medical purity. Finally, the advantages of the advanced approach for the modification of polymers via ozonation can be formulated as follows: this approach allows preparation of hydrophilic mesoporous membranes based on commercial high-tonnage and low-price polyethylene using a facile, ecologically safe, one-pot, and low-cost procedure; the ozonated mesoporous membrane materials based on polyethylene are characterized by high wettability and water permeability and can be classified as hydrophilic (contact angle well below 90°). This approach makes it possible to broaden the scope of practical applications of the ozonated polymeric membranes (cutoff 100–300kDa) for membrane filtration (nanofiltration) of small-sized objects (including viruses, proteins, foreign species, big-sized bacteria, etc.) in aqueous solutions. This approach allows repeated sterilization of membranes via ozonation. Finally, in contrast to hydrophilic commercial membranes, the resultant hydrophilic porous membranes with nanoscale pore dimensions are robust materials with high mechanical properties.

## Figures and Tables

**Figure 1 membranes-12-00733-f001:**
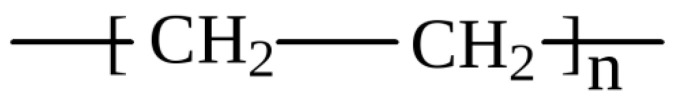
Structural formula of HDPE.

**Figure 2 membranes-12-00733-f002:**
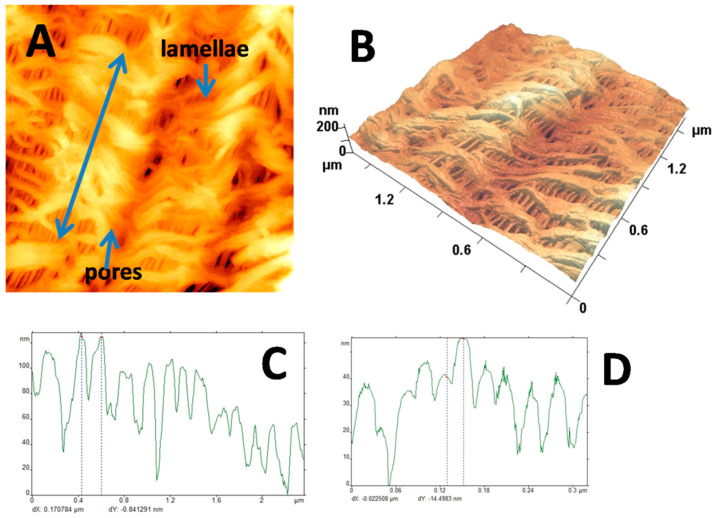
(**A**) AFM micrograph of mesoporous HDPE membrane, (**B**) 3D images of the surface relief, surface profiles of the membrane (**C**) along the direction of tensile drawing and (**D**) in the perpendicular direction (the direction of tensile drawing upon intercrystalline crazing is shown by arrow).

**Figure 3 membranes-12-00733-f003:**
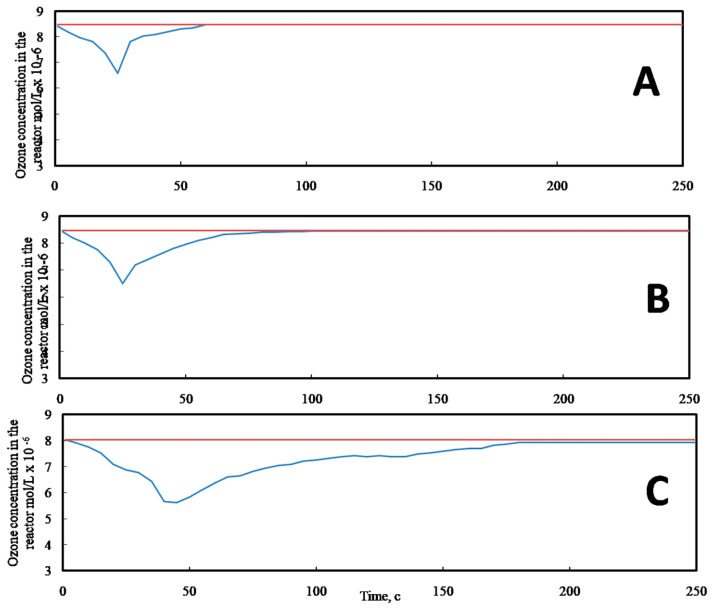
Ozone uptake curves for MPHDPE membranes before and after ozonation: (**A**) blank test without the sample, (**B**) pristine HDPE, and (**C**) mesoporous HDPE membrane (orange line—concentration of ozone in the in-flow; blue line—concentration of ozone in the off-gas).

**Figure 4 membranes-12-00733-f004:**
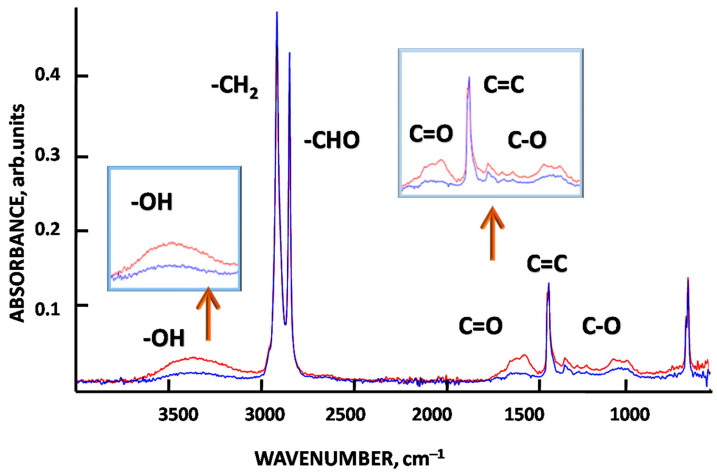
FTIR spectra of pristine HDPE (blue line) and ozonated HDPE membrane (600 s) (red line).

**Figure 5 membranes-12-00733-f005:**
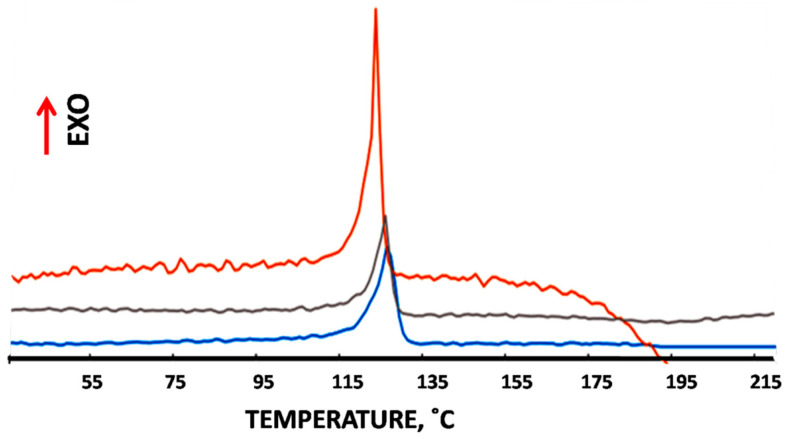
The DSC curves for mesoporous HDPE membrane (blue) and MP HDPE membranes after ozonation for 600 s (grey line) and 25,200 s (orange line). Red arrow shows the direction of exothermic effect.

**Figure 6 membranes-12-00733-f006:**
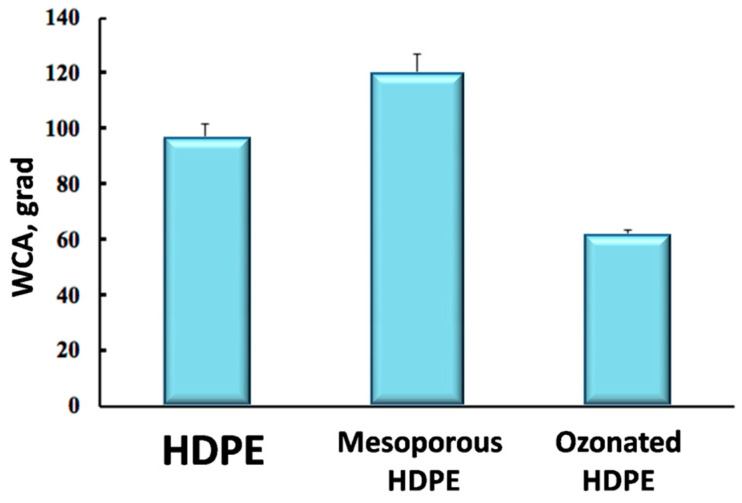
The diagram illustrating changes in WCA for pristine HDPE, MP HDPE membranes, and ozonated HDPE membranes (exposure 600 s).

**Figure 7 membranes-12-00733-f007:**
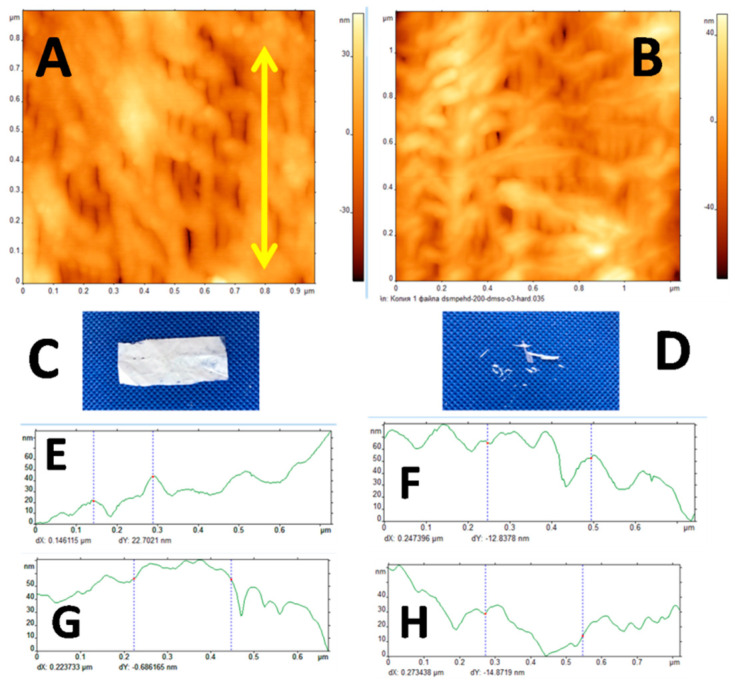
(**A**,**B**) AFM micrographs, (**C**,**D**) snapshots, and (**E**–**H**) profilogramsof the HDPE membranes after (**A**,**C**,**E**,**G**) “mild” (600 s) and (**B**,**D**,**F**,**H**) “severe” ozonation (25,200 s); profilograms (**E**,**F**) along the direction of tensile drawing and (**G**,**H**) in the transverse direction (direction of tensile drawing is shown by the arrow).

**Table 1 membranes-12-00733-t001:** Ozonation of pristine HDPE and mesoporous HDPE membranes.

Sample	Time of Ozonation, s	Ozone Uptake, L (±0.05 × 10^−9^ L)
Pristine HDPE	250	0.8 × 10^−9^
Mesoporous HDPE	250	4.2 × 10^−9^
Pristine HDPE	600	0.9 × 10^−9^
Mesoporous HDPE	600	5.6 × 10^−9^

**Table 2 membranes-12-00733-t002:** Melting temperature, heat of fusion, and degree of crystallinity for ozonated MP HDPE membranes.

Time of Ozonation, s	Melting Temperature, *T_m_*, °C	Heat of Fusion, Δ*H*, J/g	Degree of Crystallinity *χ*, %
0	126.6	169.5	57.8
600	126.0	144.5	49.3
25,200	123.7	265.1	90.47

Note: *χ*—degree of crystallinity [Δ ± 2.5%], ∆*H*—heat of fusion [Δ ± 2.5%], *T_m_*—melting temperature [Δ ± 2%].

**Table 3 membranes-12-00733-t003:** Results of the mechanical analysis.

Sample	Time of Ozonation, s	Tensile Strength, MPa (±0.02 MPa)	Elongation at Break, % (±0.2%)
Mesoporous HDPE	0	6.8	220.3
Ozonated mesoporous HDPE	600	8.4	180.6

## Data Availability

Data is contained within the article. The datasets analyzed during the current study are available from the corresponding author upon reasonable request.

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
