# Peer review of "Hydrophilization of Hydrophobic Mesoporous High-Density Polyethylene Membranes via Ozonation"

_membranes, 2022, doi:10.3390/membranes12080733_

Round 1

Reviewer 1 Report

ReportonpaperwitID-membranes-1796642

This paper discusses the hydrophilization of strongly hydrophobic, high-density polyethylene membrane via ozonation. The aim of this chemical process was to make it water permeable. Then the structure of the treated membrane was studied by different methods as e.g. DSC, AFM, FITR, and also was measured its performance. This latter was demonstrated with a single data.

The discussion of the measured results, the structure of the text is also acceptable. Against that there is an important insufficiency in this manuscript, namely the practical application(s) of the treated membrane is missing. Without it this paper is a half work to my opinion. Accordingly, I recommend to complete this manuscript with additional chapters, which are involved as:

1.      the proof of the advantageous application possibilities of the treated membrane. This statement may be confirmed by own experimental results;

2.      to show the superiority of the treated membrane, comparing its performance to other hydrophilic membranes.

According to the reviewer’s opinion, this manuscript needs major revision before its publication.

Author Response

  1. the proof of the advantageous application possibilities of the treated membrane. This statement may be confirmed by own experimental results;

ANSWER

We revised the text in Conclusions and highlighted the advantages of the treated membranes.

  1. to show the superiority of the treated membrane, comparing its performance to other hydrophilic membranes.

ANSWER

The advantages of the advanced approach for the modification of polymers via ozonation and resultant membrane materials can be formulated as follows: (1) this approach allows preparation of hydrophilic mesoporous membranes based on commercial high-tonnage and low-price polyethylene via a facile, ecologically safe, and low-cost procedure; (2) the ozonated mesoporous membrane materials based on polyethylene are characterized high wettability and water permeability, (3) this approach makes it possible to broaden the spectrum of the practical application of the ozonated membranes (cutoff 100-300 kDa) for membrane filtration (nanofiltration) of small-sized objects (including viruses, proteins, big-sized bacteria, etc.); (4) possibilities of the repeated sterilization of membranes via ozonation, (4) low-price and robust membrane materials; (5) high mechanical properties of the highly porous membranes with nanoscale pore dimensions.

Reviewer 2 Report

1. Introduction: Authors reviewed that the ozonation is a common method for surface modification. However, it is more important to propose the scientific problems in the ozonation area.

2. 2.3.2 Ozonation: The specific equipment and model should be mentioned in the Experimental section.

3. P179: What's the meaning of "achieves 50 vol.%"?

4. P212: The authors compared the surface roughness of the MP membranes and the pristine HDPE. However, the surface roughness of the pristine HDPE has not been shown in the manuscript.

5. 3.3.1: There is no obvious difference between the pristine HDPE and the ozonated HDPE membrane in Figure 4. What kind of information can be found from the FTIR spectra? 

6. P264-265: "The FTIR spectra show the presence of new functionalities in MP HDPE membrane", What kind of new groups can be found from the FTIR spectra?

7. P291-293: The authors proposed that the sample after ozonation for 25200 s has a crystallinity degree of 90.47%, which should be further confirmed by XRD.

Author Response

  1. Introduction: Authors reviewed that the ozonation is a common method for surface modification. However, it is more important to propose the scientific problems in the ozonation area.

ANSWER

Thank you for this valuable comment. We fully agree that ozonation, in addition to the solution of practical issues, offers many scientific challenges which are central to each academic research. We revised the text (INTRODUCTION).

  1. 2.3.2 Ozonation: The specific equipment and model should be mentioned in the Experimental section.

ANSWER

We used the following equipment for ozonation: flow reactor produced by Medozone, Moscow, Russia. This information is involves in Experimental section.

  1. P179: What's the meaning of "achieves 50 vol.%"?

ANSWER

When the initial non-porous HDPE films are stretched in physically active liquid environments, their deformation proceeds via the mechanism of intercrystallite crazing which involves separation of crystalline lamellae and fibrillation of the amorphous phase in the intercrysallite regions. This process is accompanied by the development of microscopic porosity. We showed that the porosity of the HDPE samples increases with increasing tensile strain and achieves 50 vol. % at a tensile strain of 200%. For our studies on ozonation, we selected the HDPE porous samples with a porosity of 50 vol %.

The text is revised as

Upon tensile drawing in n-decane, porosity W of the samples gradually increases with increasing tensile strain ε, and at ε = 200%, W = 50 vol.%.

  1. P212: The authors compared the surface roughness of the MP membranes and the pristine HDPE. However, the surface roughness of the pristine HDPE has not been shown in the manuscript.

ANSWER

We reported that, according to the analysis of the AFM images, surface roughness of the MP membranes is ~6 times higher than that of the pristine HDPE. Surface roughness was estimated using the FemtoScan Online scientific SPM image processing software. Surface roughness of pristine HDPE is 1 (arb.units), and surface roughness of the porous HDPE is 5.6.

  1. 3.3.1: There is no obvious difference between the pristine HDPE and the ozonated HDPE membrane in Figure 4. What kind of information can be found from the FTIR spectra? 

ANSWER

We revised the FTIR spectra and now we present new FTIR spectra with the better resolution (Figure 4).

  1. P264-265: "The FTIR spectra show the presence of new functionalities in MP HDPE membrane", What kind of new groups can be found from the FTIR spectra?

ANSWER

We revised the FTIR spectra and now we present new FTIR spectra with better resolution (Figure 4).

  1. P291-293: The authors proposed that the sample after ozonation for 25200 s has a crystallinity degree of 90.47%, which should be further confirmed by XRD.

ANSWER

The samples after prolog ozonation lose their integrity and stability and can be easily disintegrated into powder (Figure 7D). These materials cannot be used as membranes but this observation has triggered our attention as ozonation can offer new advantages for degradation of polymers and their utilization, and this route seems to be advantageous from the ecological viewpoint as the powerful instrument for waste treatment. Our tests on differential scanning calorimetry (DSC), which is the most traditional and reliable method for the estimation of the degree of crystallinity, showed that that value is high. However, we skip the XRD measurements as these materials can hardly be classified as membranes. In our future studies, we would like to study ozonation as the method for degradation of polymers, and we will certainly pay a special attention to the phase composition of the degraded polymers.  

Reviewer 3 Report

In this manuscript, the authors reported surface modification of mesoporous HDPE membranes via ozonation, as the result, moderate treatment led to a more hydrophilic surface that enables water permeation for a membrane once impermeable to water at low pressure. The authors evidenced that surface became smoother physically and the composition changed chemically upon ozone treatment. Overall, I found some spots in the manuscript that I cannot get behind. I would not recommend this paper for publication until the following issues are addressed:

1.      I am not convinced that the FTIR spectra showed the results of chemical composition change at the surface from Figure 4. The authors claim the band corresponding to C=O at 1715 cm-1 indicated the modification. It seems to me the change is minimal. Perhaps, the authors should give a more detailed explanation of the FTIR and including the FTIR of the HDPE undergone “severe” ozonation might help.

2.      Line 297, the authors mentioned that “the optimal duration of ozonation is below 600s”. I am curious on what led the authors to draw such conclusion. I understand that prolonged ozone treatment would destroy the membrane, however, there are too many times between 600s and 25200s. I failed to find the reason that the optimal duration must be below 600s, not between 600s and 25200s.

Small technical issues:

3.      For equation (3), there are two Tm at both sides of the equation. What are they exactly?

4.      Line 183, what does termomechanical mean? Is it supposed to be thermomechanical?

Author Response

  1.  I am not convinced that the FTIR spectra showed the results of chemical composition change at the surface from Figure 4. The authors claim the band corresponding to C=O at 1715 cm-1 indicated the modification. It seems to me the change is minimal. Perhaps, the authors should give a more detailed explanation of the FTIR and including the FTIR of the HDPE undergone “severe” ozonation might help.

ANSWER

We revised the FTIR spectra and now we present new FTIR spectra with better resolution (Figure 4).

  1. Line 297, the authors mentioned that “the optimal duration of ozonation is below 600s”. I am curious on what led the authors to draw such conclusion. I understand that prolonged ozone treatment would destroy the membrane, however, there are too many times between 600s and 25200s. I failed to find the reason that the optimal duration must be below 600s, not between 600s and 25200s.

ANSWER

We agree that our wording was far being good. The text is revised. We mean that 600 s is the sufficient duration of ozonation when the membranes become hydrophilic and preserve their integrity and show good mechanical properties.

The revised text appears as

Hence, the sufficient duration of ozonation is 600 s when the ozonated membranes acquire the hydrophilicity (water flow, low contact angle, reduced surface roughness) but preserve their integrity and show good mechanical properties.

Small technical issues:

  1. For equation (3), there are two Tm at both sides of the equation. What are they exactly?

ANSWER

We are sorry for our lack of attention. Now, the equation is corrected and reads as

Lamellar thickness was calculated according to [29]:

                   (3)

where T0m is the equilibrium melting temperature of an ideal PE crystal [30],  is the top and bottom fold surface free energy, l is the lamellar thickness, Tm is the experimental melting temperature, and ΔH is the heat of fusion per cubic centimeter of the perfect crystal.  = 0.09 J m−2 for PE [30].

  1. Line 183, what does termomechanical mean? Is it supposed to be thermomechanical?

ANSWER

Thank you for the comment. We corrected our misprint. Of course, we mean “thermomechanical tests”.

Round 2

Reviewer 1 Report

Authors did not reflect to the reviewer's remark, on the practical applications. I recommend to complete the revised version by 2 practical examples, which prove the superiority  of the treated membrane.

Author Response

Round #2

Authors did not reflect to the reviewer's remark, on the practical applications. I recommend to complete the revised version by 2 practical examples, which prove the superiority  of the treated membrane.

ANSWER

We didn’t ignore your remark and we agree that practical examples could supplement the manuscript. In the revised text (Round #1), we highlighted the application possibilities and advantages of the modified membranes. However, we would like to mention that the primary goal or the article was concerned with hydrophilization of highly hydrophobic membranes by ozonation and we proved (in our opinion) that this concept works. In your comments (Round #1), you wrote “the proof of the advantageous application possibilities of the treated membrane. This statement MAY BE confirmed by own experimental results”. We experimentally proved that the water entry threshold is reduced from 250 bar down to 1 bar, and this statement means that the ozonated membranes can be used for the nanofiltration of aqueous solutions and water treatment. We introduced the data on the cutoff which serves as guidelines for the practical use of membranes. We revised the text and showed the advantages of the modified membranes. This article was totally devoted to the feasibility of surface modification and hydrophilization of hydrophobic membranes, and the areas of their practical application were highlighted as the lighthouse for our further research activities.

The revised text appears as

Finally, the advantages of the advanced approach for the modification of polymers via ozonation can be formulated as follows: this approach allows preparation of hydrophilic mesoporous membranes based on commercial high-tonnage and low-price polyethylene using a facile, ecologically safe, one-pot, and low-cost procedure; the ozonated mesoporous membrane materials based on polyethylene are characterized high wettability and water permeability and can be classified as hydrophilic (contact angle well below 90Ëš). This approach makes it possible to broaden the scope of practical applications of the ozonated polymeric membranes (cutoff 100-300 kDa): for membrane filtration (nanofiltration) of small-sized objects (including viruses, foreign species, proteins, big-sized bacteria, etc.) in aqueous solutions. This approach allows repeated sterilization of membranes via ozonation. Finally, in contrast to hydrophilic commercial membranes, the resultant hydrophilic porous membranes with nanoscale pore dimensions. are robust materials with high mechanical properties.

Reviewer 2 Report

The figures in this paper should be modified and showed clearly. For example, Figure 4 and Figure 5 should be revised by the specific software. 

Author Response

Round #2

The figures in this paper should be modified and showed clearly. For example, Figure 4 and Figure 5 should be revised by the specific software. 

ANSWER

Thank you very much for the good advice. We revised the figures and added the revised figures to the text (marked by color).

Reviewer 3 Report

The author has addressed my concern, especially the FTIR. I recommend this manuscript for publication.

Author Response

Round #2

The author has addressed my concern, especially the FTIR. I recommend this manuscript for publication.

ANSWER

We are very grateful for your comments.